# Profiling Novel Alternative Splicing within Multiple Tissues Provides Useful Insights into Porcine Genome Annotation

**DOI:** 10.3390/genes11121405

**Published:** 2020-11-26

**Authors:** Wen Feng, Pengju Zhao, Xianrui Zheng, Zhengzheng Hu, Jianfeng Liu

**Affiliations:** Key Laboratory of Animal Genetics, Breeding and Reproduction, Ministry of Agriculture, College of Animal Science and Technology, China Agricultural University, Beijing 100193, China; wfeng@cau.edu.cn (W.F.); zhaopengju2014@gmail.com (P.Z.); zxr07sk1@163.com (X.Z.); zhengzheng0517@163.com (Z.H.)

**Keywords:** alternative splicing, transcript, protein, domain, single nucleotide polymorphism

## Abstract

Alternative splicing (AS) is a process during gene expression that results in a single gene coding for different protein variants. AS contributes to transcriptome and proteome diversity. In order to characterize AS in pigs, genome-wide transcripts and AS events were detected using RNA sequencing of 34 different tissues in Duroc pigs. In total, 138,403 AS events and 29,270 expressed genes were identified. An alternative donor site was the most common AS form and accounted for 44% of the total AS events. The percentage of the other three AS forms (exon skipping, alternative acceptor site, and intron retention) was approximately 19%. The results showed that the most common AS events involving alternative donor sites could produce different transcripts or proteins that affect the biological processes. The expression of genes with tissue-specific AS events showed that gene functions were consistent with tissue functions. AS increased proteome diversity and resulted in novel proteins that gained or lost important functional domains. In summary, these findings extend porcine genome annotation and highlight roles that AS could play in determining tissue identity.

## 1. Introduction

Alternative splicing (AS) is a regulated process that generates multiple transcripts from a single gene. It therefore plays an important role in expanding protein diversity. AS affects 95% of multi-exonic genes in humans, and occurs in high proportions in other animals [1,2,3]. AS has four basic modes, including exon skipping, alternative donor sites, alternative acceptor sites, and intron retention (IR) [4]. In addition to these, multiple promoters [5] and multiple polyadenylation sites are two other mechanisms by which different mRNAs may be generated from the same gene [6].

Many alternatively spliced isoforms play important roles in the biological timing and development of tissues [7,8,9,10]. It is becoming clear that a large number of AS events contributes to the acquisition of adult tissue functions and identity in human tissue development [11]. It has been revealed that RNA mis-splicing underlies a growing number of human diseases [12]. Protein coding genes always have several alternatively spliced isoforms, emphasizing the importance of AS in gene expression [13]. Proteins generated by different AS also have the potential to gain or lose domains. This can substantially change the protein products, which results in similar or even opposite biological functions, which in turn can affect phenotypes [14]. Point mutations, such as single nucleotide polymorphisms (SNPs), have been shown to have substantial phenotypic variation and affect pre-mRNA splicing [15,16,17,18,19,20]. AS leads to the early termination of translation by the introducing premature stop codons [21]. Regulation of AS plays roles in multiple eukaryotic biological processes, including cell growth [22], chromatin modification [23], and tissue development [24].

As the best medical models for human diseases, pigs have similar anatomical and physiological structures as humans [25]. AS in pigs plays an essential role in the regulation of gene expression in genital tissues [26,27]. However, compared to the relevant studies in humans, it has still left a gap in our knowledge of the features of pig AS in multiple pig tissues. Various research groups have attempted to understand the role of AS in pigs by using expressed sequence tags (ESTs). A total of 1223 genes, with an average of 2.8 splicing variants per gene, have been detected among 16,540 unique genes using the EST data [28]. However, it has been reported that more AS could be detected using RNA sequencing (RNA-seq) than via EST technology in humans and plants [1,29]. However, to date, only one study has cataloged AS in a cross-breed at the pig genome-wide level, using RNA-seq [30]. In this previous study by Beiki et al., they mainly focus on transcripts and transcript structures. In our study, we further detect what affects the generation of AS, and how AS affects downstream progress.

In this study, a profile of AS events within multiple tissues of the Duroc pigs was constructed to extend the porcine AS genome annotation. Genome-wide transcript identification was performed in 34 normal tissues of the pig, using over 340 Gb of sequence data generated from 116 RNA sequencing libraries. In total, 2,486,239 known transcripts and 2,430,911 novel transcripts were identified. Tissue-specific expression patterns of novel and known transcripts were examined, and then the effects of the divergence of novel isoforms on their translation were analyzed. AS variations regulated by SNPs were also analyzed. The data reported here provide a valuable resource for enhancing the understanding and utilization of pig AS.

## 2. Materials and Methods

### 2.1. Sample Collection and Sequencing

PBMCs (peripheral blood mononuclear cells) and 33 different tissues were removed from nine unrelated, healthy Duroc pigs from Shenzhen Jinxinnong Technology Co., Ltd. (Shenzhen, China) in this study (Appendix A). These nine Durocs consist of three infant pigs at the age of 3 days, three adult pigs at the age of 3 months, and three adult pigs at the age of 1 year old. For tissues collected from infant pigs, the samples are referred to tissuename_I (e.g., Brain_I), while from adult pigs they are called tissuename_A (e.g., Brain_A). The sample collection and treatment were fully conducted in strict accordance with the protocol approved by the Institutional Animal Care and Use Committee (IACUC) of China Agricultural University (no. DK2017/163) and Shenzhen Jinxinnong Technology Co., Ltd. The tissues and cells were then used for RNA and protein extraction.

### 2.2. Separation of RNA from Tissues

According to the standard protocols of Trizol method (Invitrogen, Carlsbad, CA, United States), the total RNA was isolated from mixture of equally unrelated pig pool tissues. RNA degradation and contamination were monitored on 1% agarose gels. The purity and contamination of total RNA was measured using a NanoPhotometer trophotometer (IMPLEN, Westlake Village, CA, United States) and Qubit RNA Assay Kit in a Qubit 2.0 Flurometer (Life Technologies, Carlsbad, CA, United States). The RNA’s integrity was measured using the RNA Nano 6000 Assay Kit of the Bioanalyzer 2100 system (Agilent Technologies, Santa Clara, CA, USA). RNA samples that met the criteria of having an RNA integrity number (RIN) value of 7.0 or higher and a total RNA amount of 5 μg or higher were included and batched for RNA sequencing.

### 2.3. Library Construction and RNA Sequencing

The total RNA of the samples meeting the quality control (QC) criteria were ribosomal RNA-depleted and depleted QC, using the RiboMinus Eukaryote System v2 (Thermo Fisher Scientific, Waltham, MA, USA) and RNA 6000 Pico chip (Agilent Technologies) according to the manufacturer’s protocol. RNA sequencing libraries were constructed using the NEBNext Ultra RNA Library Prep Kit (Illumina, Santiago, CA, United States), with 3 μg rRNA-depleted RNA, according to the manufacturer’s recommendation. RNA-seq library preparations were clustered on a cBot Cluster Generation System using a HiSeq PE Cluster Kit v4 cBot (Illumina), and sequenced using the Illumina Hiseq 2500 platform according to the manufacturer’s instructions, with a data size of per sample of a minimum of 10 G clean reads (corresponding to 126 bp paired-end reads). The sequenced RNA-Seq raw data for 34 pig tissues is available from NCBI Sequences Read Archive with the BioProject number PRJNA392949.

### 2.4. Read Alignment to the Reference Sus Scrofa 11.1 Genome

The RNA-Seq raw data were trimmed based on the quality control for downstream analyses by following steps: BBmap [31] automatically detected the adapter sequence of reads and removed those reads containing Illumina adapters; the Q20, Q30, and GC content of the clean data were calculated by FASTQC [32] for quality control and filtering; homopolymer trimming to 3′ end of fragments and removal of the N bases of 3′ end were carried out by Fastx toolkit v0.014 [33]. The resulting sequences then were mapped to a reference *Sus scrofa* 11.1 genome by Hisat 0.1.6-beta 64-bit [34]. Ensemble *Sus scrofa* 11.1 version 91 [35] annotation was used as the transcript model reference for the alignment and splice junction findings, as well as for all protein-coding genes and isoform expression-level quantifications. In addition, StringTie 1.0.4 [36] calculated the FPKM (fragments per kilobase of exon model per million mapped reads (controlling for fragment length and sequencing depth)) values. A gene or transcript was defined as expressed when its expression was measured above 0.1 FPKM in all tissues [37].

### 2.5. Alternative Splicing Events in Pig Transcriptome

Asprofile (v1.0.4) software (https://ccb.jhu.edu/software/ASprofile/) was used to classify and count the AS events in each sample [38]. Asprofile counts twelve AS events types in total. In our analysis, only exon skipping, alternative donor site, alternative acceptor site, and intron retention were included. Tissue-specific AS events in pig transcriptome were detected by Splicing Express software [39].

### 2.6. Comparison of Novel and Known Protein Domains

Translated DNA sequences of novel transcripts were aligned to UniProt database by DIAMOND software (v4.4.0) (https://www.crystalimpact.com/diamond/) [39]. HMMER3 [40] was used to determine conserved protein domains for novel isoforms and their most similar known transcripts. The domain changes in novel and known proteins were carried out in a pairwise manner. Protein conformation was predicted by the Swiss model [41].

### 2.7. SNP Compared with Alternative Splicing Variations

Here we mainly used Pvaas software [42] to detect the single nucleotide1 variant (SNV) mutation associated with the novel AS, which is mainly to determine the correlation between the SNV and AS by Fisher exact test. Its reliability was assessed by the *p*-value after correcting by the Benjamini–Hochberg procedure. Further filtering was performed in order to ensure the accuracy of SNV: the minor mutation frequency should be greater than 5% (the minor mutation reads number accounts for more than 5% of AS); the number of mutant reads is ≥5; the *p*-value after correction <0.001; and the reads number of the new AS is ≥10.

### 2.8. Known and Novel AS Validation Using Quantitative Reverse Transcription PCR

RNAs from six pig tissues, including heart, liver, lung, kidney, brain, and lymph were transcribed into cDNA using PrimeScript RT reagent Kit with gDNA Eraser (TaKaRa Bio, Beijing, China) for PCR reaction. Two reverse-transcribed reaction systems were conducted. For reverse-transcribed reaction system I, a DNA-free master mix of each reaction was prepared, with 2.0 μL 5× gDNAEraser buffer, 1.0 μL gDNA Eraser, 7 μL RNase-free dH2O, and 1.0 μg total RNA. Reverse-transcribed reaction system I was standing for 5 min. Reverse-transcribed reaction system II consisted of 4.0 μL 5× PrimeScript Buffer 2, 1.0 μL PrimeScript RT Enzyme Mix I, 1.0 μL RT Primer Mix, 4.0 μL RNase-free dH2O, and 10.0 μL reverse-transcribed reaction system I. Reverse-transcribed reaction system II reacting was performed at 37 °C for 15 min, followed by 85 °C for 5 s.

The primers for qPCR amplification were designed by primer-blast and confirmed by Oligo 7.0. The details about the primers are listed in Appendix A. The selected internal reference gene was β-actin, which is commonly used in swine tissue as an internal reference gene. RT-qPCR was performed by the LightCycler 480 SYBR Green I Master kit (Roche Applied Sciences, Indianapolis, IN, United States). A total of 20 μL volumes consisted of 2 μL cDNA, 10 μL SYBR green master mix, 1 μL forward primer and 1 μL reverse primer (10 μmol/L), and 6 μL nuclease water. RT-qPCR conditions were as follows: pre-incubation for 5 min at 95 °C, and amplification for 40 cycles of 10 s at 95 °C, 20 s at 60 °C, and 20 s at 72 °C. After this, a high-resolution melting curve was generated, using the following protocol: 5 s at 95 °C and 1 min at 65 °C, followed by a gradual increase in temperature from 60 °C to 97 °C, using a ramp rate of 0.02 °C per second. Results were analyzed with the standard Light- Cycler 480 software, version 1.5 (Roche), using the 2-ΔΔCt method [43] to calculate the relative expression level of the target gene for each sample.

### 2.9. Protein Extraction and Western Blotting

Proteins were isolated from the heart and kidney of the Durocs. Fresh frozen tissue was thawed, cut into small pieces, and extensively washed with pre-cooled PBS (Gibco, Rockville, MD, USA). Tissues were suspended in 100 μL RIPA Lysis Buffer (Beyotime, Nanjing, China) and supplemented with 1% proteinase inhibitor (PMSF; Beyotime, Nanjing, China). The supernatant was collected and determined with a BCA (Bicinchoninic acid) assay kit (Thermo Fisher Scientific). The protein concentration of all samples were adjusted to 2 ug/ul with RIPA buffer (Beyotime, Nanjing, China). Samples containing 30 μg protein were separated on 9% sodium dodecyl sulfate–polyacrylamide gels (SDS-PAGE), and then electrotransferred onto a nitrocellulose membrane for 1 h using Bio-Rad Trans-Blot. The membrane was blocked with 5% non-fat milk in Tris-buffered saline (20 mM Tris-HCl, pH 7.6, 137 mM NaCl) containing 0.1% tris-buffered saline +Tween-20 (TBST, Gibco) for 30 min at room temperature, and incubated at 4 °C overnight with the following primary antibodies: Anti- Immunoglobulin Binding Protein 1(IGBP1) antibodies (ab70545, Abcam, Cambrige, UK). The membranes were washed three times with TBST for 10 min and incubated with goat anti-rabbit IgG (Heavy + Light) secondary antibody, an Horseradish peroxidase (HRP) conjugate (Thermo Fisher Scientific), for 1 h at room temperature. The antibody–antigen complexes were detected using Western blotting (WB) luminal reagent. The bands on the developed film were quantified with Quantity One v4.6.2 software (Bio-Rad, Hercules, CA, United States). The β-actin was used as a loading control for normalization.

## 3. Results

### 3.1. Identification of Novel Transcripts in 34 Different Pig Tissues

To discover and map novel transcripts, RNA-seq of 34 different pig tissues was performed. On average, 48.48 million reads per tissue were sequenced from 116 strand-specific and paired-end cDNA libraries (Appendix A). Of these sequences, an average of 43.97 million reads (90.7%) per sample passed the strict quality control (QC). A total of 1495 million high-quality reads (376.7 Gb, 135-fold genome coverage) were aligned to the porcine reference genome (*Sus scrofa* 11.1), and 1223 million mapped fragments (310.1 Gb, 110.8-fold genome coverage) with an average alignment rate of 88.29% were recovered (Appendix A). These mapped reads were then assembled and quantified as candidate transcripts using Stringtie software. This step produced a total of 2,486,239 transcripts from 29,270 genes across all tissues; of these, 60,578 transcripts (23,887 loci) were annotated in the pig Ensemble database (https://www.ensembl.org/), and 144,134 transcripts from 2424 non-coding genes, as well as 15,385 coding genes, were considered as potential AS. The remaining 2,281,529 novel transcripts were from 26,493 loci without any annotated information. Details for each tissue are available in Appendix A. These novel transcripts enhanced the pig genome annotation and increased the number of average transcripts for each gene in pigs.

### 3.2. Classification of Alternative Splicing Types

AS events were classified into 12 types, including the four basic types of AS events (exon skipping, alternative donor site, IR, and alternative acceptor site) (Figure 1A). A total of 138,403 AS events and 29,270 genes across all 34 tissues were detected. A gene had 4.73 AS events on an average. Alternative donor sites accounted for the most common AS event type (60,851; 44%), whereas the other three types shared similar percentages (exon skipping, 19%; alternative acceptor site, 19%; IR, 18%) (Figure 1A). The most significant increase was evident in the number of novel alternative donor sites, which accounted for 44% of all novel AS events (Figure 1B). The average length of IR was found to be the longest, with exon skipping the shortest, which is consistent with the definition of each AS type (Figure 1B).

The novel AS events were categorized into two types, novel transcripts of known genes and novel transcripts of unknown genes. The number of novel transcripts in different tissues is shown in Appendix A. More novel transcripts of unknown genes than known genes were detected. The correlation between the number of previously annotated isoforms and novel isoforms of annotated genes was analyzed (Figure 2A). Transcripts with ensemble IDs were considered to be annotated: the greater the number of annotated isoforms per gene, the more the novel isoforms were detected (Figure 2A). Comparing to annotated isoforms, novel isoforms did not increase significantly, perhaps due to the limited isoforms per gene, which would not change even with improved annotation methods.

The number of AS events related to different genes varied. The number of AS events of the AHNAK gene was 391. However, thousands of genes only had a single AS event. To explain the variation in the number of AS events of the genes, a cluster analysis was performed. Genes with more than 10 AS events were grouped together; those with between 1 and 10 events were grouped together, and those with one AS event were grouped together. A Kyoto Encyclopedia of Genes and Genomes (KEGG) analysis of these three groups (Figure 2B, Appendix A) determined that for genes with one AS event, a total of 24 pathways were significantly enriched, including the phosphatidylinositol signaling system (*p*-value = 6.30 × 10^−9^; number of genes = 59), inositol phosphate metabolism (*p*-value = 2.80 × 10^−6^; number of genes = 43), glycerophospholipid metabolism (*p*-value = 1.30 × 10^−4^; number of genes = 49). These genes were mostly involved in lipid metabolism. Other substantial pathways were mainly related to cancer and reproduction. For genes with 1 < AS events ≤ 10, altogether 70 pathways were significantly enriched. Forty-six genes were involved in the measles pathway (*p*-value = 3.9 × 10^−8^; number of genes = 46). Genes with 1 < AS events ≤ 10 were mostly involved in diseases and signaling pathways, such as the forkhead box O (FoxO) and hypoxia-inducible factors (HIF)-1 signaling pathways. However, for genes with >10 AS events, only seven pathways were significantly enriched. These pathways were related to neuroactive and immune-related diseases. The second significant pathway of the seven was the olfactory transduction pathway (*p*-value = 4.9 × 10^−145^), enriched with 639 genes.

### 3.3. Tissue Specificity of Alternative Splicing in Different Tissues

The tissue-specific AS events in the pig transcriptome were detected with Jekroll splicing express [39]. As reported in a previous study [37], transcripts with an expression level above 0.1 FPKM in only one tissue and an expression level less than 0.1 FPKM in all other tissues were defined to be tissue-specific transcripts. The number of tissue-specific transcripts varied substantially in different tissues, with the peripheral blood mononuclear cells possessing the most unique isoforms (1633) and the pancreas possessing the least (133; Table 1). Further examination of these tissue-specific transcripts showed that 86% of the genes that encode them were expressed in a single tissue at levels above 0.1 FPKM, revealing that the tissue specificity of these transcripts occurs at the transcriptional level. The remaining 14% of the tissue-specific transcripts had other isoforms expressed in other tissues, indicating that their tissue specificity likely stems from AS (Figure 3A).

To identify developmentally regulated changes in AS, StringTie [36] was used to quantify the expression of novel and known transcripts in different tissues (Appendix A). The average level of expression of novel transcripts in the pancreas, liver, longissimus dorsi, spleen, prostate, adrenal gland, and breast was higher (FPKM > 10) than that of known transcripts. The difference in the expression level of transcripts between the highest expression transcript and the lowest expression transcript was compared (the ratio of major FPKM and minor FPKM) to the gene whose number of AS events was greater than 1. This revealed that more isoforms of a gene are associated with a greater difference in its highest and lowest expression (Figure 3B). A total of 116,439 genes were tissue-specific, of which 109,773 genes were newly detected. A total of 5,683 genes were expressed across all 34 tissues, suggesting that many genes are expressed in multiple tissues simultaneously (Appendix A). These results are useful resources for improving the transcript annotations of the pig genome.

### 3.4. Potential Effect of Novel Alternative Splicing on the Pig Proteome

Translated DNA sequences of novel transcripts were aligned to the UniProt database using DIAMOND software [44]. To determine the effect of AS on the pig proteome, computationally predicted proteins encoded by the novel isoforms of known genes were compared to their respective known annotated proteins. A total of 1184 alternatively spliced transcripts were mapped to 361 new proteins (Appendix A). These results, therefore, improved the annotation of unknown proteins in the pig genome.

HMMER3 (biosequence analysis using profile hidden Markov models) was used to identify conserved domains within the novel and known proteins, which were then compared in a pairwise manner [40]. A novel transcript of apolipoprotein B MRNA editing enzyme catalytic subunit 3B (APOBEC3B) in uterine tissue was mapped to a protein with the UniProt identifier D3U1S2_PIG. Compared to a similar known transcript (UniProt identifier: F1SNY0_PIG), the novel transcript (3072 bp) was 11 bp shorter. The annotated protein contained two APOBEC-like N-terminal domains, whereas the new isoform lacked an APOBEC-like N-terminal domain, potentially altering its function substantially (Figure 4A). The novel transcript had three coding exons, which was fewer than the known transcript (Figure 4B). It has been shown that DNA ligase 3 (LIG3) proteins activate leukemia via a transcriptional error [45]. The current results reveal a novel transcript (UniProt identifier: I3L9T2_PIG) of LIG3 that had gained an additional domain (DNA ligase 3 BRCA1 C-terminal domain) not present in the known transcript (UniProt identifier: I3LN18_PIG; Figure 4C). This novel transcript had one more coding exon than the known transcript (Figure 4D). The protein conformations of these four transcripts were predicted (Figure 4E,F), and the loss of the APOBEC3B domain and the gain of the LIG3 domain were clearly demonstrated. Although AS may cause changes in protein conformation, further experiments are needed to understand expression and function of the predicted proteins.

### 3.5. Conserved Alternative Splicing between Pig and Human

Pigs are generally considered a promising medical model for human diseases, and pig orthologs for many human disease-associated genes have been identified [25]. To investigate the orthologous relationship between humans and pigs at the protein level, pig proteins were aligned with human proteins using Basic Local Alignment Search Tool (BLAST; https://blast.ncbi.nlm.nih.gov/), with the criteria of minimum length ≥ 50 bp and identity ≥ 80%. As such, 4168 (56.97%) pig proteins that are orthologous to human proteins were identified.

A similar study reported a global analysis of AS transitions between human infants and adult hearts [46]. Infant and adult pig hearts were included in the current study. Similar to human infant hearts, IR and exon skipping occurred more frequently in the infant pig. In contrast, exon skipping occurred more frequently in the adult pig hearts, but least frequently in the adult human hearts. The specific AS genes that were differentially expressed in infant human and pig hearts were compared. Two genes, sperm-associated antigen 5 (SPAG5) and LIG3, were found to be shared between human and pig. Pig proteins of SPAG5 and LIG3 were aligned to their human proteins (Figure 5A). For two proteins, the identity between the pig and human was 100% and 99.5%. These conserved proteins indicate genome evolution and facilitate further exploration of the potential functional similarity between human and pig genomes.

### 3.6. Effect of Single Nucleotide Polymorphism on Pig Alternative Splicing

Deep RNA sequencing can detect sequence variations associated with AS. AS events were grouped into three types: canonical splicing with a GT–AG intron (i.e., GT and AG splicing signals at donor and acceptor sites, respectively), semi-canonical splicing with GC–AG or AT–AC introns, and novel splicing without GT–AG introns [47]. Semi-canonical and novel splicing are also called aberrant splicing. SNPs occur in different regions of a gene. They change the expression of transcripts and therefore the protein abundance, especially when present in the exonic regions (Appendix A). The changes in exons owning to AS may cause synonymous or non-synonymous mutation in the proteins. Some aberrant splicing of the exons led to non-synonymous mutations in the codons, and caused the premature termination of translation in some tissues (Appendix A). Related SNPs, genes, and codons when an aberrant splicing occurred in more than 20 tissues are described in Table 2. Six novel splicing events were shown to cause non-synonymous mutations in genes (Table 2)—in particular, a missense SNV (exon8:c.G970T:p.D295Y) within the synapse-associated protein 1 (SYAP1) gene was found that generated a serine to threonine substitution (Figure 5B), leading to a change in the SYAP1 protein conformation (Figure 5C).

### 3.7. Validation of Known and Novel Transcripts Using RT-qPCR and Western Blotting

The reliability of transcript identification in this study was verified with RT-qPCR of one known and five novel transcripts from six tissues. Of these transcripts, TCONS_01698330 and TCONS_01698333 are known and novel transcripts of DDX17 gene, respectively. TCONS_00393548, TCONS_00028775, TCONS_01245051, and TCONS_01413087 are novel transcripts of MICU2, IGBP1, TCIRG1, and NFATC2IP genes, respectively. These six transcripts that were detected with RT-qPCR and RNA-seq showed consistent expression patterns (Figure 6A). Gene expression levels measured using these two methods were highly correlated (*R*^2^ = 0.92). These results confirm the accuracy of the identification and characterization of pig transcripts.

To determine whether AS was also reflected at the protein level, Western blotting was performed. Among the five genes used for RT-qPCR, only the antibody of IGBP1 (Abcam) for pig was available. β-actin was used as house-keeping gene. Fortunately, only one AS type and transcript of the IGBP1 gene was detected. We can see that the mRNA of IGBP1 is translated to similar changes at protein levels.

## 4. Discussion

In the present study, novel AS from 34 different tissues in nine Duroc pigs was profiled via deep RNA sequencing, providing important insights into porcine genome annotation. A total of 138,403 AS events and 29,270 expressed genes were detected, with 4.73 AS events per gene on average. The average number of AS events per gene has improved to 1.93 splicing variants in this study compared with the 2.8 splicing variants per gene by EST data [28,30]. Previous studies have shown that exon skipping, alternative donor sites, alternative acceptor splice sites, and IR are the four basic AS types. In the current study, the alternative donor site was the most common AS form, accounting for 44% of the total AS events. The percentage of the other three AS forms was approximately 19%. These results suggest that AS events involving different alternative splicing types can produce different transcripts or proteins that affect biological processes. In a previous study to detect AS events in nine pig tissues using Iso-seq and RNA-seq data, alternative acceptor splice site was shown to be the most prevalent AS type in each tissue [30]. The study by Beiki et al. [30] shared four tissues in common with the current study, including the brain, liver, spleen, and thymus. In the current study, exon skipping was the predominant splicing event in the brain, liver, and spleen (Table 1), which is consistent with a study where exon skipping is the most prevalent AS type in animals [4]. In addition, IR was the most frequent of AS events in the thymus, which is similar to that reported in humans [48]. Nine Duroc pigs, three infants and six adults, were used to detect AS events in the present study, whereas a single cross-bred pig was used for AS analysis in the previous study [30]. Compared to results in humans, we can see it is common that AS show different trends in different situation. Even we do not share the exact same results, both of our results make sense. The different breeds used in these two studies could have produced different results, but regardless of the splicing mechanism, this large increase in new, alternatively spliced transcripts improves our knowledge of the diversity of the pig proteome dramatically.

Genes with only one AS event were predominantly located in the pathway of lipid metabolism. Genes with more than 10 AS events were associated with pregnancy or disease-related signaling pathways. Whether genes belonging to a pathway are related in terms of their AS events requires further investigation. The functions of genes with tissue-specific AS were found to be nearly consistent with their tissue of expression. In addition, the expression of a transcript in different tissues could be different. For example, the expression of the gene glutathione peroxidase 4 (GPX4) ranges from 20 to over 400 FPKM, and its transcript expression in the adult testes was much higher than in other tissues (Appendix A). The GPX4 gene translates into specific enzymes in the testes of adult rats, which is an important structural protein in mature sperm [49,50].

Genes expressed in specific tissues are reported as tissue-enriched genes with tissue-specific functions in multicellular organisms (Appendix A). Results shows that 8482 well-annotated protein isoforms demonstrated tissue-specific expression characteristics. The expression level of 16,356 well-annotated protein isoforms in a specific tissue is at least five times higher FPKM compared to its second-highest expression level in the tissue. In the present study, tissues with complex biological processes usually had more tissue-enriched genes that are closely related to the function of the corresponding tissue. For example, the rhodopsin (RHO) gene, enriched in the retina, plays an important role in the deposition of retinal pigments [51]. Therefore, these specific, tissue-enriched genes can not only confirm the biological properties of known genes, but also predict the potential function of unknown genes in the pig genome. Besides these tissue-specific genes, there are still some genes that can be highly expressed in some functionally related tissues/organs, and these genes usually have similar functions. A total of 1318 well-annotated protein isoforms could be divided into seven types (Appendix A). These results show that most of the enriched genes belong to the brain system (72.7%), followed by the muscle system, adrenal and thymic system (6.6%), and liver and gallbladder system (4.5%).

In general, the majority of novel splicing events resulted in new proteins. Novel alternatively spliced transcripts often contain domain losses or gains relative to their most similar known isoform, and even led to opposite functions [14]. In the current analysis, proteins encoded by novel transcripts were predicted in silico. The 1184 alternatively spliced transcripts mapped to 361 new proteins. Novel and known transcripts of APOBEC3B and LIG3 were compared as examples and showed changes in transcript length. The novel transcripts of APOBEC3B and LIG3 showed the loss and gain of a domain, respectively, as predicted by HMMER3 (Figure 4A,D). However, additional experiments are still necessary to confirm the expression and function of new proteins.

SNPs affect protein-binding sites (or cause mutations in the binding proteins themselves) and contribute to aberrant splicing. Some non-synonymous mutations also cause premature translational termination. A missense SNV (exon3:c.T376A:p.S126T) within SYAP1 was found that generated a serine to threonine substitution, which caused changes in protein conformation of these two isoforms, as predicted by the Swiss model.

## 5. Conclusions

In summary, the current analysis greatly expands the pig transcriptome to include more than 2,281,529 newly annotated transcripts. It was found that if genes had more AS events, the expression difference between the highest and lowest expressions would be greater. The expression of genes with tissue-specific AS events is consistent with their tissue-specific functions. This expansion of isoforms increases the known proteome diversity and results in novel proteins that gain and lose important functional domains. Point mutations, such as SNPs, in exons could lead to new AS. Taken together, these findings extend pig genome annotation and highlight the roles that AS plays in tissue identity in organisms.

## Figures and Tables

**Figure 1 genes-11-01405-f001:**
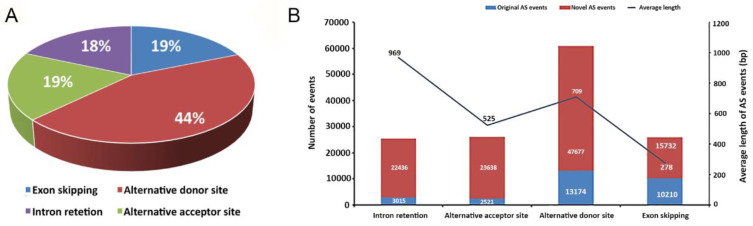
Four modes of alternative splicing (AS). (**A**) The proportion of exon skipping, alternative donor sites, alternative acceptor sites, and intron retention. (**B**) Left *y*-axis: the number of known (blue bars) and novel (red bars) alternative splicing events. Right *y*-axis: average length of alternative splicing events (black polyline).

**Figure 2 genes-11-01405-f002:**
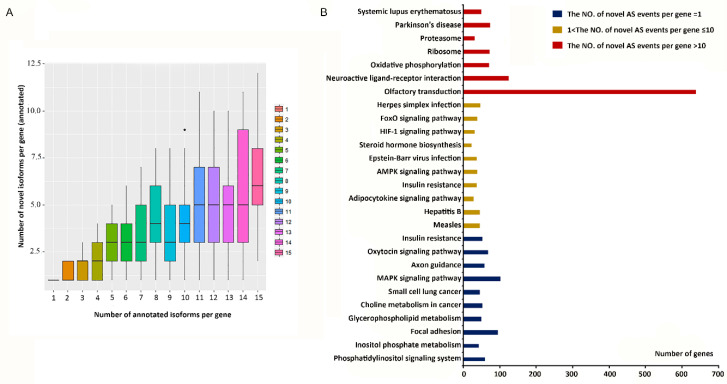
The difference in the number of alternative splicing-related genes. (**A**) The relationship between the number of annotated isoforms and novel isoforms per gene. (**B**) Top 10 enriched Kyoto Encyclopedia of Genes and Genome (KEGG) pathways of genes with different numbers of AS events (ASEs). Blue bars represent genes with one ASE, yellow bars represent genes with 1 < ASEs ≤ 10, and red bars represent genes with >10 ASEs.

**Figure 3 genes-11-01405-f003:**
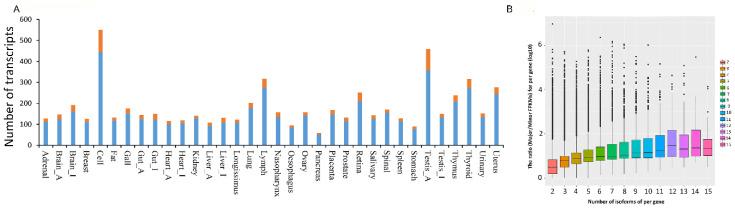
(**A**) Single tissue-specific transcripts. The number of known and novel transcripts that are expressed at or above 0.1 FPKM (fragments per kilobase of exon model per million mapped reads) in only one tissue and less than FPKM in all others. Blue areas represent the transcripts and related genes expressed only in one tissue (gene expression dependent). Red areas represent the transcripts expressed in one tissue, with other isoforms present in other tissues (alternative splicing dependent). (**B**) The expression of a gene with different numbers of alternative splicing. The *y*-axis represents the ratio = log10* (the highest FPKM/lowest FPKM) of a gene with multiple AS events, which means the greater the ratio, the greater the difference between the highest and lowest expression of a different transcript per gene. The plot represents this ratio for a gene.

**Figure 4 genes-11-01405-f004:**
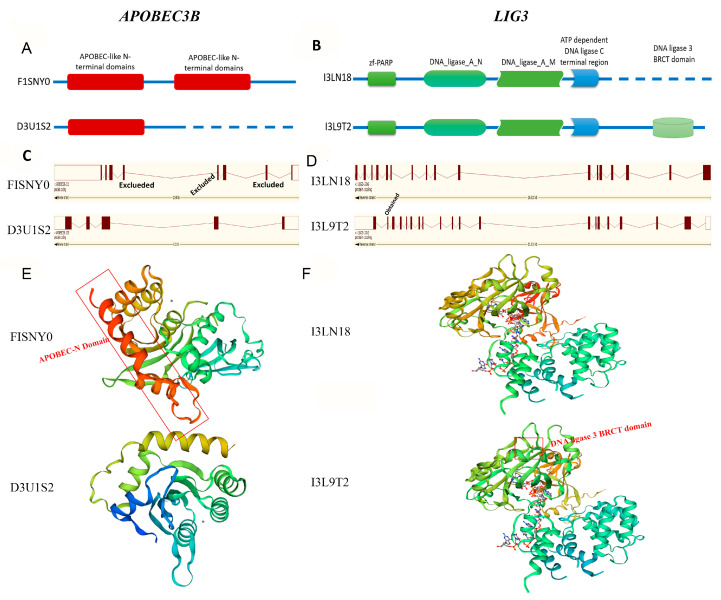
Examples of proteins encoded by known transcripts compared with those encoded by novel transcripts. (**A**) Apolipoprotein B MRNA editing enzyme catalytic subunit 3b (APOBEC3B)’s loss of an APOBEC-like N-terminal domain. (**B**) DNA ligase 3 (LIG3)’s gain of a DNA ligase 3 BRCA1 C-terminal (BRCT) domain. (**C**,**D**) The number of coding exons of APOBEC3B and LIG3. (**E**,**F**) The predicted protein conformations of APOBEC3B and LIG3.

**Figure 5 genes-11-01405-f005:**
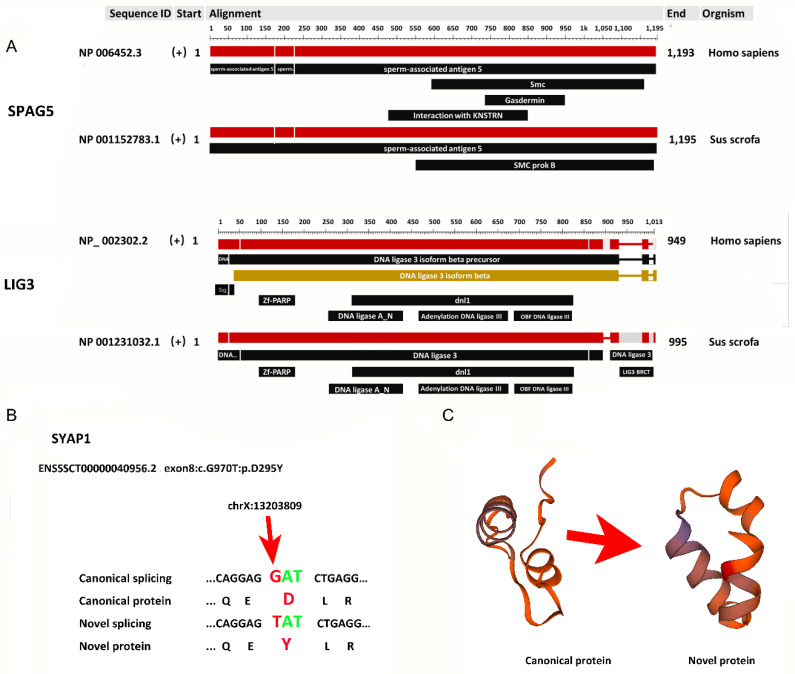
(**A**) Example of a homologous protein related to an AS gene between human and pig (downloaded from NCBI Multiple Sequence Alignment Viewer, Version 1.15.0). (**B**) Effect of single nucleotide polymorphism (SNP) mutation of synapse-associated protein 1 (SYAP1) on alternative splicing. (**C**) Protein conformation of canonical and novel isoforms as predicted by the Swiss model.

**Figure 6 genes-11-01405-f006:**
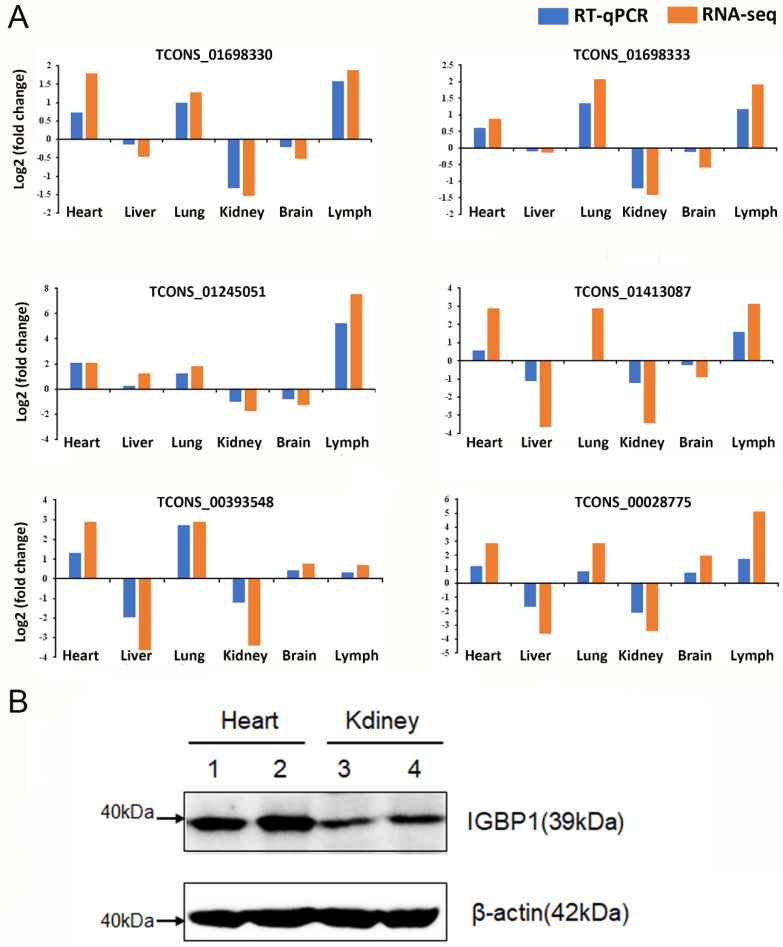
(**A**) Validation of transcripts using RT-qPCR. Blue bars and red bars represent gene expression measured with RT-qPCR and RNA-seq, respectively. (**B**) The protein level of Immunoglobulin Binding Protein 1 (IGBP) using western blotting.

**Table 1 genes-11-01405-t001:** Distribution of tissue ASE types in different tissues.

Tissue	AS with Gene	Exon Skipping	Alternative 5′ Splice Site	Alternative 3′ Splice Site	Intron Retention
Brain_A	147	113	60	52	58
Breast	126	92	80	71	63
Testis_I	150	80	71	70	99
Lung	202	94	95	75	225
Placenta	167	141	102	83	99
Stomach	89	48	56	48	34
Gut_A	144	76	111	54	86
PBMC	551	402	303	212	716
Fat	132	78	72	63	81
Lymph	317	133	497	116	479
Prostate	132	61	58	75	105
Thymus	238	215	194	118	224
Heart_A	116	89	55	72	56
Brain_I	192	101	54	81	116
Gall	175	78	70	64	146
Nasopharynx	157	84	87	82	92
Retina	252	192	113	104	228
Thyroid	316	114	82	79	350
Liver_A	108	97	69	66	45
Gut_I	150	77	69	77	107
Oesophagus	95	86	63	32	57
Salivary	143	67	128	48	117
Urinary	152	70	40	61	136
Testis_A	459	359	322	172	313
Heart_I	119	102	38	52	56
Kidney	140	156	113	88	78
Ovary	157	91	69	65	127
Spinal	170	84	84	80	102
Uterus	276	102	62	101	286
Adrenal	127	99	56	49	84
Liver_I	131	131	85	69	68
Longissimus	122	91	90	53	58
Pancreas	58	41	33	19	40
Spleen	128	117	56	64	75

**Table 2 genes-11-01405-t002:** Statistics of different SNP mutation types in different tissues.

Position	Type	Canonical	Aberrant	Gene	Mutation	Detail
chr7: 22889860	Novel	T28228	T2132	TMP-CH242-74M17.2	Non-synonymous	ENSSSCT00000001330
		C188	C17845	ENSSSCG00000001229		exon3:c.A395G:p.H132R
chr7: 22889835	Novel	A27034	A1996	TMP-CH242-74M17.2	Non-synonymous	ENSSSCT00000001329
		C0	C2429	ENSSSCG00000001229		exon3:c.T370G
chr7: 22825399	Novel	T2495	T2956	SLA-1	Non-synonymous	ENSSSCT00000036412
		C0	C2075	ENSSSCG00000001231		exon3:c.A404G:p.D135G
chr1: 236442561	Novel	G2635	G1216	TLN1	Non-synonymous	ENSSSCT00000005856
		T0	T722	ENSSSCG00000005317		exon8:c.C745A:p.H249N
chr9: 67827403	Novel	G1128	G7	C4BPA	Non-synonymous	ENSSSCT00000017061
		T593	T7237	ENSSSCG00000015663		exon4:c.C605T:p.T202I
chr9: 67827383	Novel	G1506	G16	C4BPA	Synonymous	ENSSSCT00000017061
		A520	A8310	ENSSSCG00000015663		exon4:c.G585A:p.K195K
chr4: 90572872	Semi-canonical	C17145	C27	TAGLN2	Synonymous	ENSSSCT00000007008
		T7513	T785	ENSSSCG00000006395		exon3:c.G246A:p.G82G
chr7: 1955628	Semi-canonical	A6919	A26	TUBB2B	Synonymous	ENSSSCT00000001098
		G5	G4623	ENSSSCG00000001006		exon1:c.T42C:p.N14N
chr6: 103414552	Novel	A6090	A79	LOC733637	Synonymous	ENSSSCT00000025362
		G14	G325	ENSSSCG00000003692		exon4:c.T495C:p.F165F
chrX: 13203809	Novel	T21856	G0	SYAP1	Non-synonymous	ENSSSCT00000040956.2
		G0	T4538	ENSSSCG00000012148		exon8:c.G970T:p.D295Y
chr7: 24913889	Semi-canonical	C17224	C14	SLA-DRB1	Synonymous	ENSSSCT00000001612
		T61	T1190	ENSSSCG00000001455		exon1:c.G39A:p.A13A
chr6: 103414681	Novel	A19996	A114	LOC733637	Synonymous	ENSSSCT00000025362
		G481	G332	ENSSSCG00000003692		exon4:c.T366C:p.G122G
chr7: 23636509	Novel	C9015	C3066	SLA-7	Synonymous	ENSSSCT00000035331
		T2779	T2870	ENSSSCG00000024161		exon4:c.C852T:p.H284H

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
