# Peer review of "Profiling Novel Alternative Splicing within Multiple Tissues Provides Useful Insights into Porcine Genome Annotation"

_genes, 2020, doi:10.3390/genes11121405_

Round 1

Reviewer 1 Report

I did arranje some time to check the authors reply to my comments and verified that they revised the manuscript accordingly.

Author Response

Thanks for your kindly suggestion. We have checked our paper again and made our utmost efforts modify the grammar mistake.

Reviewer 2 Report

The authors have addressed my previous concerns and improved their manuscript.

I have a minor comment: Since the protein expression and function cannot be tested (due to the lack of specific antibodies), to prevent misleading the readers, the authors need to include a caveat in section 3.4 that this analysis cannot accurately predict the expression and function of the predicted proteins in cells or the authors could say that additional experiments are needed to understand expression and function of the predicted proteins. 

Author Response

Q:I have a minor comment: Since the protein expression and function cannot be tested (due to the lack of specific antibodies), to prevent misleading the readers, the authors need to include a caveat in section 3.4 that this analysis cannot accurately predict the expression and function of the predicted proteins in cells or the authors could say that additional experiments are needed to understand expression and function of the predicted proteins. 

Answer:Thanks for your kindly remarks. We have added two statements in the section 3.4 and Discussion.

3.4. Potential Effect of Novel Alternative Splicing on the Pig Proteome

...Although AS may cause changes in protein conformation, further experiments are needed to understand expression and function of the predicted proteins.

Discussion

...

The novel transcripts of APOBEC3B and LIG3 showed loss and gain of a domain, respectively, as predicted by HMMER3 (Figure 4A & 4D). But still, additional experiments are necessary to confirm expression and function of new proteins.

...